# Testicular Germ Cell Tumours—The Role of Conventional Ultrasound

**DOI:** 10.3390/cancers14163882

**Published:** 2022-08-11

**Authors:** Jane Belfield, Charlotte Findlay-Line

**Affiliations:** Liverpool University Hospitals NHS Foundation Trust, Liverpool L7 8XP, UK

**Keywords:** ultrasound, testis, seminoma, teratoma, germ-cell, tumour

## Abstract

**Simple Summary:**

Testicular tumours are the most common cancer found in young men, and germ cell tumours account for the majority. This article describes the use of conventional ultrasound in germ cell tumours and how to perform the ultrasound, including how to improve patient position and scanning technique. The classification of germ cell tumours is also provided, and the salient features of different types of tumour are described. Some pitfalls and difficulties of ultrasound are discussed, as well as other uses of ultrasound in this group of patients.

**Abstract:**

Testicular tumours are the most common tumours found in young males and germ cell tumours account for 95% of testicular tumours. Ultrasound is the first-line radiological investigation for imaging of the testis. This article outlines how to undertake an ultrasound examination, including optimal patient position, scanning technique and imaging parameters. Classification of germ cell tumours is provided, and salient imaging features of different tumours are described. Difficulties and pitfalls of ultrasound are described, including tumours found after a trauma presentation, orchitis causing diagnostic difficulties and imaging of small testicular lesions. Other uses of ultrasound are outlined, including looking for a primary testicular tumour following the discovery of retroperitoneal lymph nodes, imaging when tumour-makers increase, local recurrence in the scrotum, and for solid organ biopsy in metastatic disease. Conclusion: Ultrasound remains the first-line of investigation for imaging of the testis, and conventional ultrasound still plays a large role in imaging. On ultrasound alone, accurate morphological characterisation of tumours remains a challenge, despite the imaging features that can be seen in different tumour types. Therefore, histology following orchidectomy of a germ cell tumour remains the gold standard for accurate tumour characterisation.

## 1. Introduction

Testicular tumours are the most common tumours in young males, and there is a peak incidence between 25 and 34 years [1]. Germ cell tumours (GCT) account for 95% of testicular tumours, with the remainder being due to lymphoma and other less common pathologies. Testicular lymphoma usually presents in males over 60 years old, being the most common testicular malignancy in this age group [2]. The overall incidence of germ cell tumours is increasing [3], but the exact reasons for this are uncertain.

Several risk factors are known, including previous undescended testis, orchidopexy, family history, previous contralateral testicular tumour, and syndromes, including Klinefelter syndrome and dysplastic nevus syndrome. Previous studies suggest that 5–20% of patients with a previous history of an undescended testis will develop a tumour in the contralateral testis [4], and having an undescended testis increases the risk of developing a germ cell tumour by 5–10 times [5].

Testicular microlithiasis was previously thought to be a risk factor, but newer evidence has shown that it is only a risk factor for a testicular tumour in the presence of another independent risk factor. Ultrasound is useful for surveillance in this group of patients and is recommended annually until the age of 55 years, following which regular self-examination is advised [6]. If testicular microlithiasis is seen during an ultrasound examination, the patient should be questioned to ascertain if they have any other independent risk factors for germ cell tumours [6].

Ultrasound (US) is the first-line radiological examination for imaging of the testes due to its high sensitivity, patient acceptance and relatively low cost [7]. Ultrasound has been shown to have a sensitivity and specificity greater than 90% for detecting a testicular malignancy [8]. Patients often present with a palpable mass, which is usually painless, but are reported to present with pain in up to 15% of cases [9]. Most palpable lumps felt by the patient are benign and due to other pathologies, including epididymal cysts, hydroceles, scrotal calculi, or testicular cysts [10].

Ultrasound is very useful in distinguishing intratesticular from extratesticular lesions. Ultrasound is able to confirm the presence of an intratesticular mass and is mainly used for identification and initial characterisation. Extratesticular lesions are much less commonly malignant in nature than an intratesticular lesion [11], with approximately 3% of extratesticular solid lesions being reported as malignant [12].

The differentiation of different types of germ cell tumour is challenging using ultrasound, and recent research suggests that tumour morphological features can be suggestive of a particular type of germ cell tumour [13]. Additionally, tumour markers such as lactate dehydrogenase (LDH), beta human chorionic gonadotropin (Beta–HCG) or alpha-fetoprotein can also suggest the subtype of germ cell tumour. However, histology from an orchidectomy specimen remains the gold standard examination for accurate characterisation of the tumour.

Recent advances in ultrasound include the use of elastography and contrast-enhanced ultrasound.

This narrative review focuses on the use of conventional ultrasound in germ cell tumours, and features as a part of a special edition on testicular tumour imaging for this journal.

This is an important topic, as conventional ultrasound is used routinely as the first-line investigation for suspected testicular malignancy; therefore, a good understanding of how to perform and interpret ultrasound is needed.

The database PubMed was used to search for previously published articles using relevant key words such as ‘testicular germ cell tumours’, ‘seminoma’, ‘teratoma’ ‘ultrasound’, with the applicable papers reviewed for the article. This paper is written as a narrative review in order to include both the relevant literature as well as knowledge and experience from the authors in their routine clinical practice.

## 2. How to Perform Ultrasound

Before performing the ultrasound, it is recommended that the operator questions the patient about where they have noticed symptoms, what they have felt on palpation and to find out the exact location of the abnormality. If a lesion cannot be identified during the ultrasound examination, it is useful to ask the patient to demonstrate exactly where they can feel the abnormality, and then perform the ultrasound directly in that location.

Testicular ultrasound should be performed with the patient in the supine position, with the penis placed onto the anterior abdominal wall so that the scrotum is clearly visible, and the patient is as comfortable as possible. A pad or cushion can be placed under the scrotum in order to elevate the contents and make the ultrasound assessment of all the scrotal structures easier to perform [14].

A high-frequency linear transducer (7–14 Hz) should be used, and a systematic approach is required. Ideally the gel should be warmed prior to the investigation, and enough gel should be used to ensure no gas is trapped in the skin folds of the scrotum. If gel is left on the scrotal skin and the examination is too timely, the gel can dry and become trapped in between scrotal folds and hair, causing some artefacts that can make the images difficult to interpret [15].

Both testes should be evaluated in longitudinal and transverse planes with B-mode ultrasound [16]. Colour and spectral Doppler should routinely be applied to both testes as part of the examination. The epididymis and appendages should be imaged, and an image of both testes side by side should be acquired to allow for direct comparison.

The asymptomatic side should be scanned first in order to allow the appropriate setting of the gain parameters when performing a direct comparison between the two testes. The careful examination of the asymptomatic side is particularly important if a testicular tumour is seen, to exclude a synchronous tumour, although the risk of simultaneous bilateral tumours is low, between 1–5% [17]. It is important to ascertain if a mass is intratesticular or extratesticular, and whether it is cystic or solid in composition.

Any mass that is seen within the testis should be clearly examined using 2D ultrasound, and colour Doppler should be applied to assess for the vascularity. If a lesion is seen, further images should be taken and the lesion carefully evaluated to look for evidence of cystic change, haemorrhage, calcification, and to assess the overall echogenicity and margins of the lesion.

If there is a large hydrocele and the testes cannot be clearly identified, changing to a curvilinear probe is recommended as this will often help identify the testes, although the resolution is not as clear.

Images should be optimised by using the whole screen on the ultrasound machine and ensuring that the image is in the centre of the screen. Labelling of the images should be performed prior to saving any image, and it is imperative to ensure that the correct side marker is in place.

The ultrasound report should clearly state the side of the lesion, along with specific imaging features, including size, location within the testis, vascularity, presence of calcification, solid and cystic components. The appearance of the contralateral testis should also be documented, including the presence of microlithiasis.

## 3. Classification of Germ Cell Tumours

Germ cell tumours originate from spermatogenic cells and account for 90–95% of primary testicular tumours [18,19], where approximately 50% are seminomas and 50% are histologically more diverse non-seminomatous tumours (NSGCT*)* [18].

Non-seminomatous lesions include pure embryonal carcinoma, teratoma, yolk sac tumour and choriocarcinoma. Tumours are often a combination of multiple cell types, termed mixed germ cell tumours.

In 2016, the WHO updated its classification system of germ cell tumours. It introduced a new title for the precursor lesion: germ cell neoplasm in situ (GCNIS) [20]. Tumours are further subdivided into GCNIS-related tumours and non-GCNIS related tumours based on the presence or absence of chromosome 12p amplification [20].

The accurate diagnosis of the tumour type has significant implications both prognostically and therapeutically but can be difficult to ascertain using ultrasound. Studies have been conducted to predict the type and size of testicular tumours, and there are some features that are more suggestive of seminomatous tumours compared with NSGCT [21].

## 4. Imaging Features of Germ Cell Tumours

### 4.1. Seminomatous Germ Cell Tumour

Seminomas are the most common type of germ cell tumour, comprising 40–50% of GCTs, with a peak incidence between 35–45 years [22]. At presentation, the majority are limited to the testis but approximately 20% of patients will have retroperitoneal lymph node involvement with 5% also having distant metastases [19]. They are associated with a high serum lactate dehydrogenase (LDH) level [20]. Seminoma tumour cells are relatively uniform and have a better prognosis as they are radiosensitive and chemosensitive [2,19].

On ultrasound, seminomas are usually homogeneous and of low echogenicity [20]. They may be small nodules with a well-defined margin or a large infiltrating mass that can replace the whole testis. Some studies have shown that small seminomatous tumours were uniformly solid, but larger lesions appeared more heterogeneous with areas of calcification and cystic foci [13] (Figure 1). Cystic components are reported to be seen in up to 10% [23]. They are usually seen to be vascular when colour Doppler is applied, but this may not always be seen in smaller tumours [24]. It has also been reported that seminomas can be found bilaterally in 2% of patients [24].

### 4.2. Non-Seminomatous Germ Cell Tumour (NSGCTs)

This comprises all germ cell tumours that are not seminomas, and these may be found as isolated tumours, or in combination as a mixed-germ cell tumour. The peak incidence is earlier than seminomas, between 20–35 years, where they are more likely to present with advanced disease [20,25]. On ultrasound imaging NSGCTs typically have irregular margins and a mixed echogenicity. Calcifications and cystic components are more common and reported in up to 40% of tumours [20].

### 4.3. Embryonal Cell Carcinoma

In comparison to seminomas, embryonal cell carcinomas are more aggressive [20]. These tend to be heterogeneous in echogenicity with irregular borders and may contain calcification or internal haemorrhage. They can cause a contour irregularity of the testes with the invasion of the tunica albuginea [25] (Figure 2).

### 4.4. Teratoma

Pure teratomas in post pubertal males are rare and are more commonly seen in children [25]. They are usually large at presentation and approximately a third of males will have advanced disease at presentation [18]. However, tumour markers are usually normal [20]. The ultrasound imaging features are of a predominately cystic, mixed echogenicity intratesticular mass with well-defined borders [25]. Teratomas can also contain macroscopic fat and calcifications [18] (Figure 3).

### 4.5. Yolk Sac Tumour

In the pure form, yolk sac tumours predominately affect children under the age of 2 and are rarely seen in post-pubertal males [18,25]. They are associated with significantly raised serum alpha-fetoprotein levels [20]. On ultrasound, they are usually a poorly defined solid heterogenous mass and can contain internal haemorrhage, cystic foci, necrosis and calcifications [18].

### 4.6. Choriocarcinoma

Choriocarcinomas are the most aggressive of all the NSGCTs [20]. It is common for these tumours to have an early metastatic spread to the lungs, liver, gastrointestinal tract and brain [23]. Patients usually have significantly raised serum human chorionic gonadotropin levels (Beta-HCG) [20]. Pure choriocarcinomas are rare and are more often seen as a component of mixed germ cell tumours [18] (Figure 4). Ultrasound findings correspond to a heterogenous and hypoechoic mass, where internal haemorrhagic areas are often seen [18].

## 5. Difficulties and Pitfalls

Patients may present following trauma, and a previously unknown testicular tumour may be found incidentally during the ultrasound examination. The appearance may mimic a haematoma on grey-scale ultrasound [26,27]. Vascularity assessment can be useful as tumours are often vascularised, but haematomas do not typically demonstrate internal vascularity [28] (Figure 5). However, it is not always easy to distinguish a tumour from a haematoma at the time of acute trauma. Follow up ultrasound is advised following testicular trauma if a focal intratesticular lesion is seen, in order to exclude an underlying tumour [28]. A haematoma will change in echotexture, size and shape on repeat ultrasound as the haematoma begins to liquefy and resolve, whereas a testicular tumour will remain stable when early follow up is performed.

Rupture of the testis due to an unknown testicular tumour has also been described [29]; therefore, an underlying tumour should be considered, particularly in patients presenting with mild trauma. If there is any concern, tumour markers and a repeat ultrasound are advised to ensure that there is no underlying tumour.

The testes can be hypervascular in the presence of orchitis, which can make interpretation difficult. Cases have been described in the literature where patients presented with an enlarged, heterogeneous, hypervascular testis on ultrasound and it was difficult to distinguish a tumour from orchitis on the initial ultrasound [30] (Figure 6). Tuberculosis epididymo-orchitis is a manifestation of genitourinary tuberculosis [31], which may also mimic a testicular tumour on clinical presentation [32,33], so a careful ultrasound examination is advised. This is not always possible in developing countries, where the patient may be treated based on the clinical findings alone without access to ultrasound [34]. However, where possible, ultrasound is advised as the first-line investigation for a patient presenting with either a palpable lump or symptoms suggestive of epididymo-orchitis. Again, if the diagnosis is uncertain, repeat ultrasound is advised as the findings of orchitis should resolve following treatment compared with those of a testicular tumour.

Small (<5mm) testicular lesions can be challenging as they can be seen as an incidental finding on ultrasound due to the improved resolution of ultrasound transducers [35]. Previous studies show that an incidentally found small testicular solid nodule is benign in up to 80% of cases [36] and these are often seen during infertility investigations. If a small lesion is seen in the testis on ultrasound, standard evaluation is advised (Figure 7). The lesion should be assessed to ascertain if it is solid or cystic, vascular or avascular and whether it contains internal microliths or macrocalcifications [35]. If there remains a concern, or the lesion cannot be characterised on conventional ultrasound, contrast-enhanced ultrasound or a follow-up ultrasound is advised. MRI can be useful as a problem-solving tool for assessment of testicular pathology [37].

## 6. Other Uses of Ultrasound

If a patient is found to have a retroperitoneal nodal mass on cross-sectional imaging, or a primary retroperitoneal tumour, ultrasound of both testes is advised to ensure there is no testicular tumour, including a previous burned-out tumour (Figure 8). Burned-out tumours are tumours that have previously shrunk [38] and may be seen incidentally or in the presence of retroperitoneal nodes [35].

Ultrasound should be performed prior to a retroperitoneal nodal mass being biopsied, but in clinical practice the biopsy often takes place before a testicular ultrasound has been performed.

Ultrasound can be performed in patients with a previous germ cell tumour whose tumour markers begin to increase to ensure that there is no tumour in the residual contralateral testis. The 15-year risk of developing a germ cell tumour in the contralateral testis is reported as approximately 1.2% [39]. Ultrasound can be performed annually until the age of 55 years in patients with known testicular microlithiasis who have an independent risk factor for a germ cell tumour [6].

Patients may develop local recurrence in the scrotum, which may be felt as a palpable lump, and ultrasound can be used to assess any new lesion. This may be due to direct disease recurrence or nodal disease (Figure 9). This is particularly important in patients who had a scrotal incision at the time of primary orchidectomy [40], as the nodal spread differs and spreads to local pelvic nodes rather than classic retroperitoneal nodal spread which is seen following an inguinal incision [41].

In a patient presenting with metastatic disease who needs to start a chemotherapy regime prior to primary orchidectomy, ultrasound can be useful to acquire a specimen for histology, particularly in the presence of a liver metastasis (Figure 10). Primary testicular biopsy is not usually recommended due to tumour seeding and the alteration of nodal spread [42], but biopsy from the liver or retroperitoneal nodal mass can be acquired if needed.

## 7. Discussion

Ultrasound remains the first-line investigation for a patient presenting with a palpable scrotal lump and can often provide immediate reassurance to the patients when a benign lesion, such as an epididymal cyst, is confirmed. Many departments can provide access to ultrasound without a long delay, ensuring that patients with a germ cell tumour can have the lesion identified early, hopefully before nodal or metastatic disease has developed.

B-mode and colour Doppler ultrasound remain the conventional ultrasound imaging techniques and identify the majority of intratesticular and extratesticular lesions. The majority of intratesticular lesions that demonstrate increased vascularity are malignant. There are some mimics of malignant tumours, including haematomas, infarcts, congenital adrenals rests, abscesses and epidermoid cysts, that may be mistaken for a malignant tumour on ultrasound. If there is a history of trauma or recent infection, repeat ultrasound should be recommended for further evaluation of the visualised abnormality to ensure that there is no underlying testicular tumour.

Characterising testicular lesions with ultrasound remains a challenge, particularly in NSGCT which are often of mixed cell types. There are features that can be suggestive of one type of tumour, but histology remains the gold standard in giving a formal tumour composition report. Table 1 summarises the salient features that are described for different germ cell tumours.

Higher resolution ultrasound machines may enable the further characterisation of intratesticular lesions and further studies can be performed to correlate ultrasound and histological findings. Newer ultrasound techniques, including elastography [43], contrast-enhanced ultrasound [44] and 3D imaging [45] may also help characterise these lesions in the future.

With advancing ultrasound techniques and constantly improving ultrasound equipment, morphological characterisation may be more possible in the future. However, patients with suspected testicular tumours on ultrasound are likely to undergo orchidectomy, and many departments can provide a histological report with accurate morphological information about the tumours in a timely manner, allowing suitable treatment to commence without patient harm. Characterising a tumour as mixed cell is unlikely to be possible with ultrasound alone; therefore, histological examination is likely to remain the gold standard investigation for tumour characterisation.

## 8. Conclusions

Ultrasound remains the first-line investigation for patients presenting with a suspected tumour. Identification of a tumour allows the clinicians to undertake further management, including the measurement of tumour markers, orchidectomy and staging examinations. However, morphological characterisation remains a challenge on ultrasound alone despite imaging features that can be seen in different tumour types. With advancing ultrasound techniques and improved images and equipment, morphological characterisation may continue to improve in the future and is an area where further research can be performed.

## Figures and Tables

**Figure 1 cancers-14-03882-f001:**
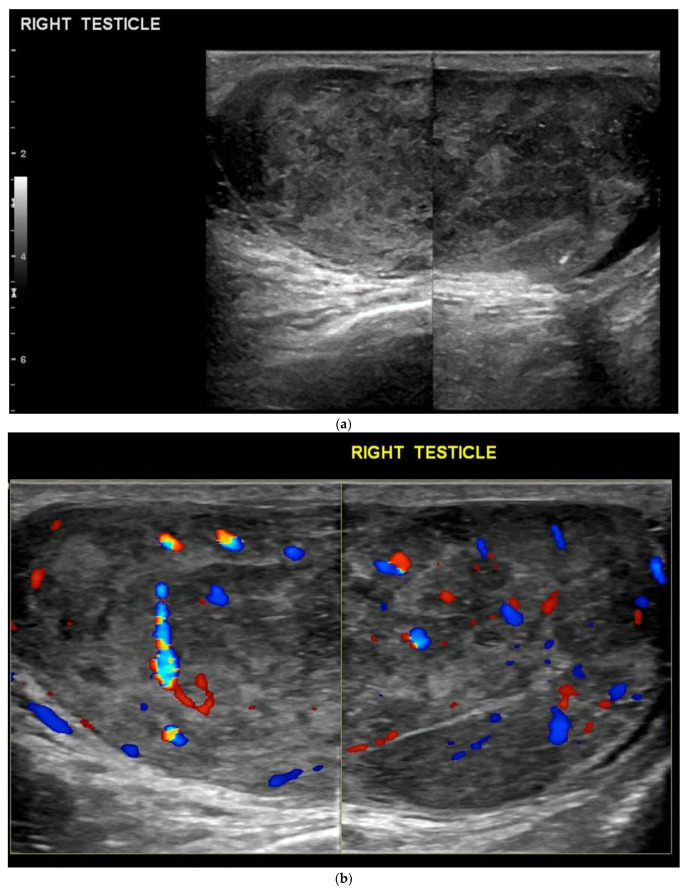
(**a**) B-mode ultrasound shows a heterogeneous large mass infiltrating and enlarging the right testis, containing areas of calcification and small cystic foci. (**b**) Colour Doppler ultrasound demonstrates increased vascularity throughout the mass.

**Figure 2 cancers-14-03882-f002:**
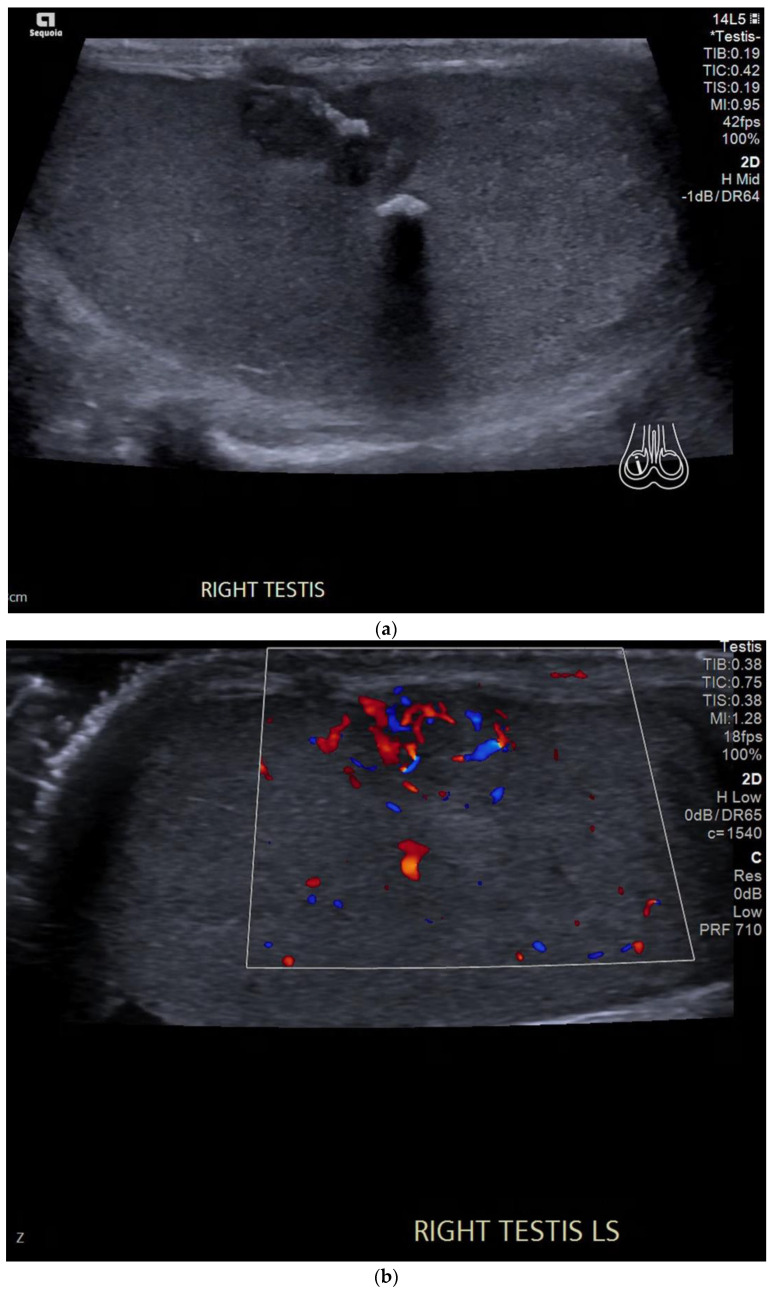
(**a**) B-mode ultrasound shows a heterogeneous lesion in the right testis containing macrocalcification and extending to the tunica albuginea and causing disruption of the testicular contour. (**b**) Colour Doppler ultrasound showing increased vascularity in the tumour extending to the tunica albuginea.

**Figure 3 cancers-14-03882-f003:**
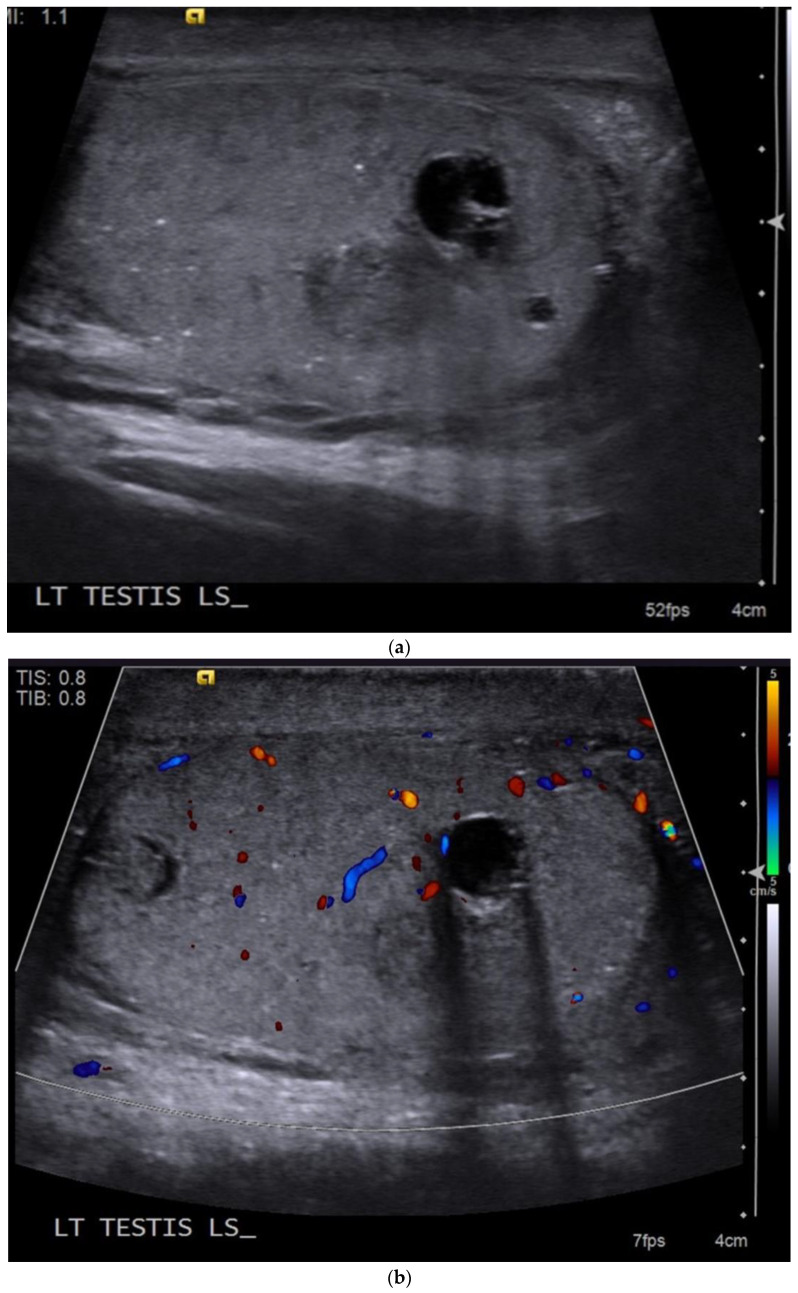
(**a**) B-mode ultrasound demonstrates well-defined intratesticular cystic and solid lesions with associated areas of calcification and additional microlithiasis. (**b**) Colour Doppler shows increased vascularity in the periphery of the tumour but not in the cystic component.

**Figure 4 cancers-14-03882-f004:**
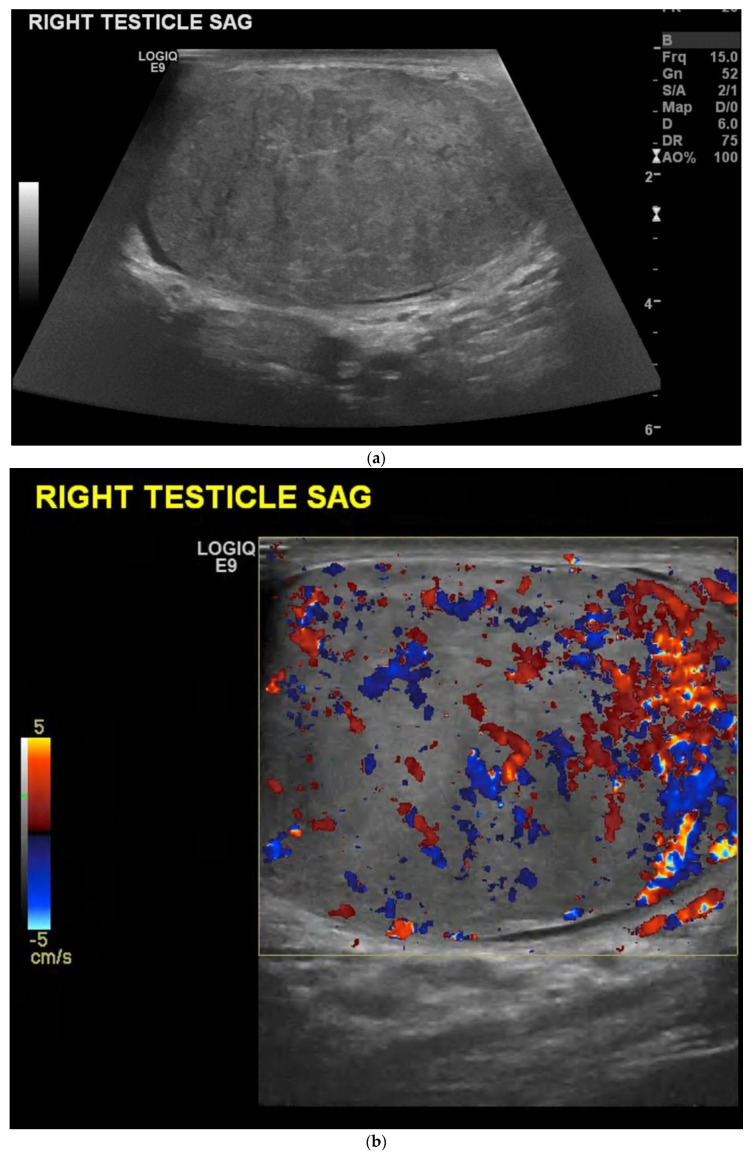
(**a**) B-mode ultrasound showing enlargement and infiltration of the testis by a large tumour, confirmed to be a mixed germ cell tumour. (**b**) Colour Doppler image shows marked vascularity throughout the lesion.

**Figure 5 cancers-14-03882-f005:**
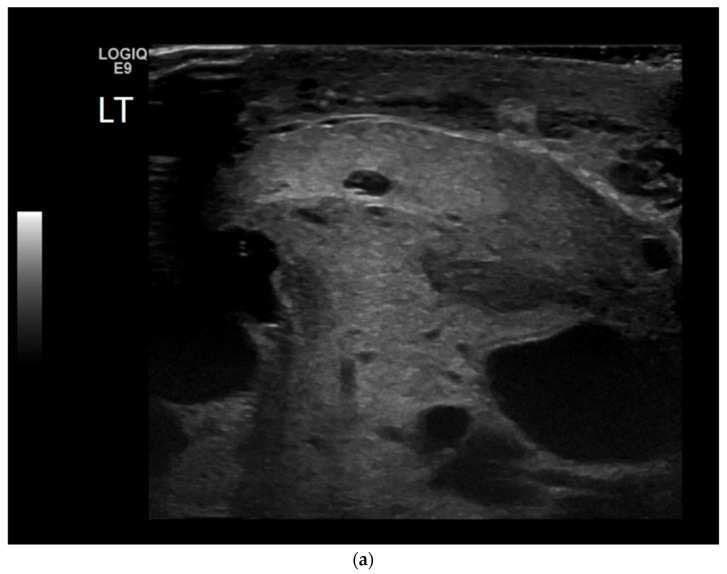
(**a**,**b**) B-mode ultrasound demonstrates thickened scrotal skin and a 5.9cm lesion with cystic and solid components in a patient following trauma. (**c**) Colour Doppler shows increased vascularity within the solid components. Subsequent orchidectomy confirmed a teratoma, which presented following trauma.

**Figure 6 cancers-14-03882-f006:**
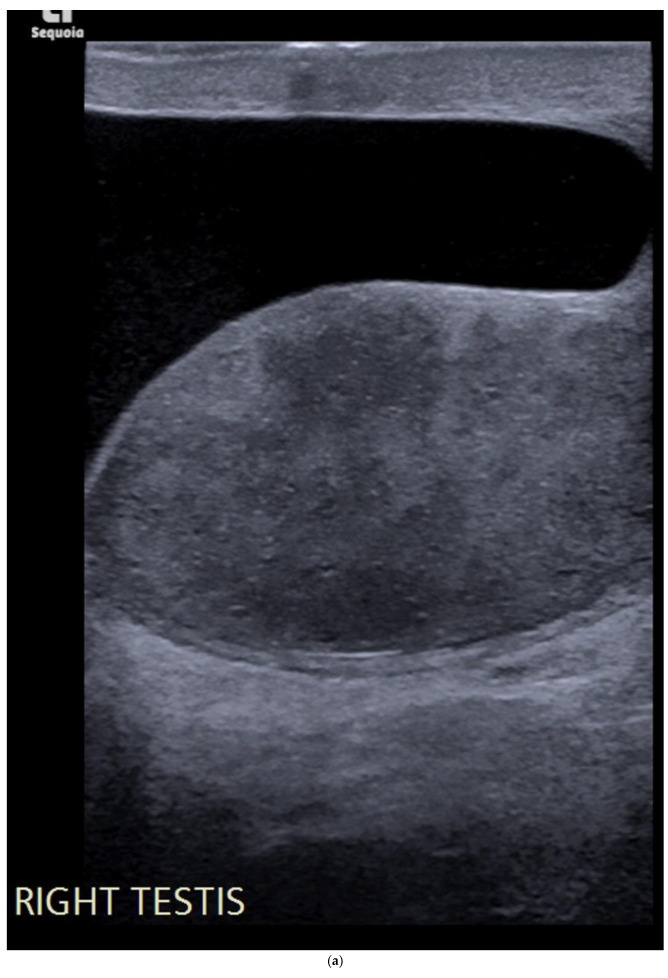
(**a**) B-mode ultrasound shows an enlarged, heterogeneous testis with a hydrocele. The epi-didymis was also enlarged, in keeping with epididymo-orchitis, but can be a tumour mimic. (**b**) Colour Doppler shows increased vascularity throughout the testis.

**Figure 7 cancers-14-03882-f007:**
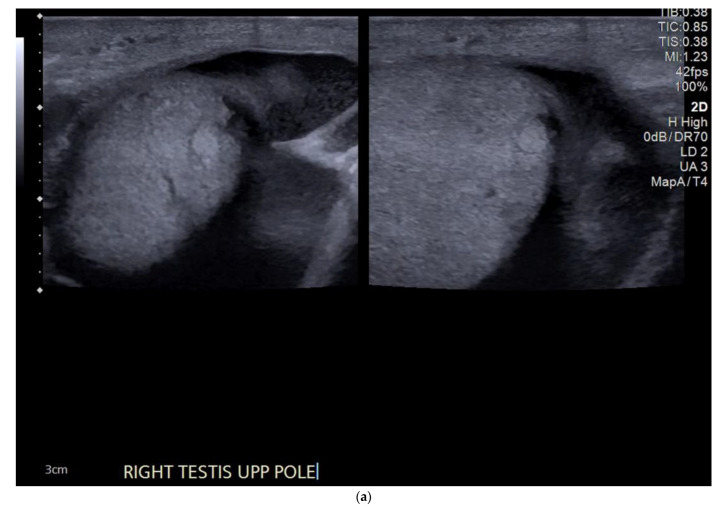
(**a**) Well-defined hyperechoic lesion in the right testis measuring < 5mm in all dimensions. (**b**) The lesion is confirmed to be avascular. Previous ultrasound two years earlier showed that the lesion was unchanged in size and appearance.

**Figure 8 cancers-14-03882-f008:**
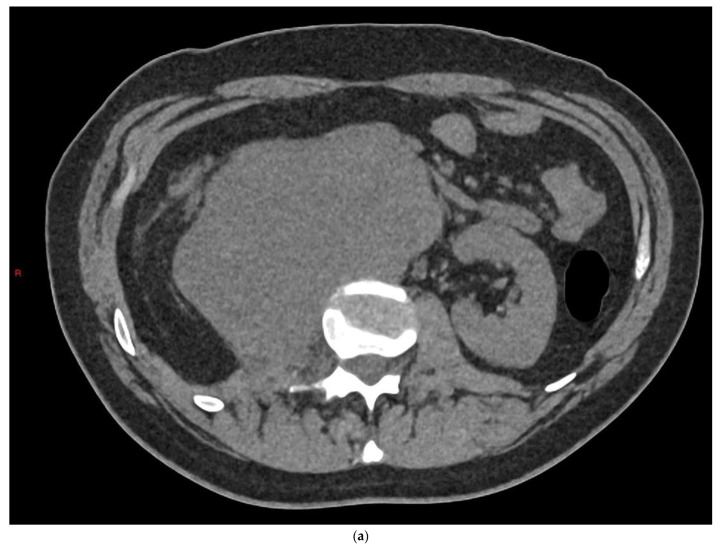
(**a**) Axial slice from CT KUB in a 35-year-old patient who presented with acute right flank pain demonstrates large retroperitoneal nodal disease. (**b**) Coronal image from contrast CT shows large nodal disease causing right hydronephrosis, deviation of the aorta and a solitary liver metastasis. (**c**) Colour Doppler ultrasound from the same patient shows a solitary lesion in the right testis in keeping with a primary testicular tumour.

**Figure 9 cancers-14-03882-f009:**
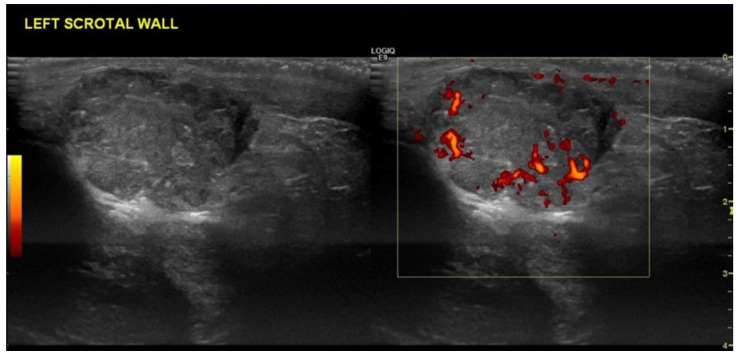
B-mode and power Doppler ultrasound in a patient with previous testicular malignancy demonstrates local recurrence in the scrotum. This patient presented following trauma and had undergone a scrotal incision rather than the conventional inguinal approach.

**Figure 10 cancers-14-03882-f010:**
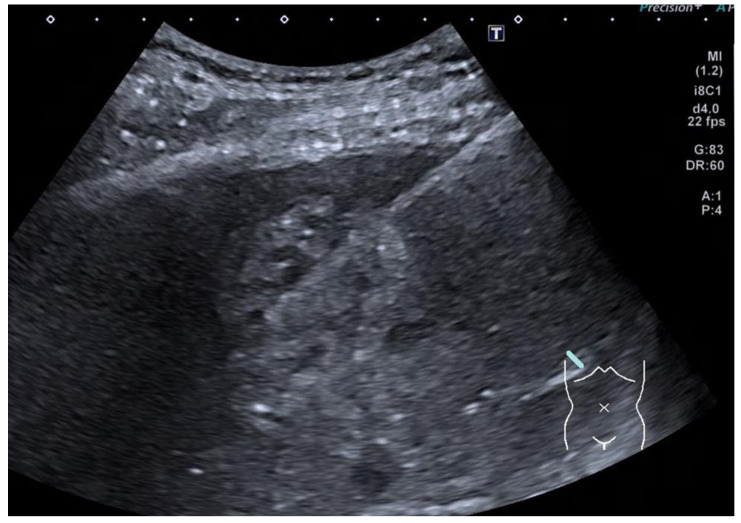
Ultrasound taken during biopsy of the solitary liver lesion, which confirms metastatic disease.

**Table 1 cancers-14-03882-t001:** Summarising ultrasound features.

Tumour Type	US Features
Seminomas	Homogenous and hypoechoic Well circumscribed Occasionally contain cystic components or calcifications
Non-seminomas	Heterogenous Irregular margins Cystic and calcification components seen commonly
Pure embryonal	Heterogeneously hypoechoic Ill-defined margins with invasion of the tunica albuginea Can contain haemorrhage and calcifications
Teratomas	Complex mass with well-defined borders Predominately cystic but can also contain macroscopic fat and calcifications
Yolk sac tumours	Poorly defined solid heterogenous mass Can contain cystic foci, internal haemorrhage, necrosis and calcifications
Choriocarcinomas	Hypoechoic and heterogenous solid mass Internal haemorrhagic areas often seen

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
