# Peer review of "Testicular Germ Cell Tumours—The Role of Conventional Ultrasound"

_cancers, 2022, doi:10.3390/cancers14163882_

Round 1
Reviewer 1 Report
Jane Belfield et al. in ” Testicular Germ Cell Tumours – The Role of Conventional Ultrasound”, show the use of conventional ultrasound in germ cell tumours and how to perform the ultrasound, including how to improve patient position and scanning technique. Some difficulties and other uses of ultrasound in this group of patients are described.
The review is well written and extremely original. Beautiful work!
Author Response
Thank you for your kind comments and review.
Reviewer 2 Report
The Authors performed a narrative review and drafted and educational piece addressing the role of conventional ultrasound imaging in the clinical setting of testicular germ cell tumor. The manuscript is generally well-written, clear and addresses the main points of the topic presented in a well-structured and timely manner. I would suggest:
1) To state the reason why this topic was addressed and why authors feel it is relevant.
2) To state the type of review performed (narrative review) and the reason for this choice.
3) I would suggest introducing a paragraph describing the methodology employed in the review. In particular:
a. Please refer to the PICOTS criteria and frame populations, intervention(s), comparator(s), outcomes, timing, setting.
b. Please provide details in the search strategy, with a search string. I would, also suggest to refer to the PRISMA criteria (please add a PRISMA flow-chart).
Author Response
To state the reason why this topic was addressed and why authors feel it is relevant.
We have added a couple of sentences in the introduction stating why the topic was addressed, as part of the special issue on testicular tumour imaging, and why a good understanding of how to perform and interpret ultrasound is needed.
To state the type of review performed (narrative review) and the reason for this choice.
We have explained that this is a narrative review, based on current literature as well as personal experience from the authors.
I would suggest introducing a paragraph describing the methodology employed in the review.
We have explained that we undertook a literature review using PubMed, and included some of the relevant key words that were used during the search.
We are unsure if PICOTS criteria of a PRISMA flowchart are applicable to the article as it has been written as a narrative review. We have provided a reasonable number of references, including guidelines, book chapters and peer-review journal publications and used these to write the narrative review.
Other similar reviews that we have read and used as references do not all include a PRISMA flowchart. However, if the reviewer thinks we need to expand further please let us know and we will endeavour to do so in due course.
Reviewer 3 Report
I congratulate with the Authors for their work, It is a rather simple but exhaustive work on ulstrasounds in germ cell tumors of the testis.
May I suggest just to check line 86 and 95 regarding the asymptomatic testis.The Authors state that the asymptomatic testis has to be initially checked (line 86) , but in line 95 they write that in case of a suspected tumor the asymtomatic testis has to be checked in detail. Please clarify. The Authors correctly mention tumor markers through the paper, but I think that a couple of lines in the introduction sunchapter may be added .
No mention on B cell lymphomas. The paper is on germ cell tumors of course, but a line on this disease may be interesting,
Author Response
Check line 86 and 95 regarding the asymptomatic testis - we have made the suggested alteration as requested, to make the important point about bilateral synchronous tumours. Hopefully the change in the order of the wording is less confusing that what was previously written.
Tumour markers - we have added a sentence in the introduction about tumours markers as requested.
Lymphoma - We have added a sentence on the demographics of testicular lymphoma but have not described ultrasound findings in detail as this is out of the remit of the paper.
Round 2
Reviewer 2 Report
I do not have any further comment.